

# Auditory interfaces in automated driving: an international survey

Pavlo Bazilinskyy and Joost de Winter

Department of Biomechanical Engineering, Faculty of Mechanical, Maritime and Materials Engineering, Delft University of Technology, Delft, The Netherlands

## ABSTRACT

This study investigated peoples' opinion on auditory interfaces in contemporary cars and their willingness to be exposed to auditory feedback in automated driving. We used an Internet-based survey to collect 1,205 responses from 91 countries. The respondents stated their attitudes towards two existing auditory driver assistance systems, a parking assistant (PA) and a forward collision warning system (FCWS), as well as towards a futuristic augmented sound system (FS) proposed for fully automated driving. The respondents were positive towards the PA and FCWS, and rated the willingness to have automated versions of these systems as 3.87 and 3.77, respectively (on a scale from 1 = disagree strongly to 5 = agree strongly). The respondents tolerated the FS (the mean willingness to use it was 3.00 on the same scale). The results showed that among the available response options, the female voice was the most preferred feedback type for takeover requests in highly automated driving, regardless of whether the respondents' country was English speaking or not. The present results could be useful for designers of automated vehicles and other stakeholders.

# INTRODUCTION

## The development of automated driving systems

The development of automated driving technology is a key topic in modern transportation research. A transition to automated driving may have a large positive influence on society (*European Commision, 2011*). Each year more than 1,000,000 fatal accidents occur on roads worldwide, with the lower-income countries being overrepresented (*Gururaj, 2008*; *World Health Organization, 2013*). If automated driving systems are designed to be fully capable and reliable, a very large portion of—yet probably not all—road traffic accidents could be prevented (*Goodall, 2014*). Furthermore, traffic congestions, gas emissions, and fuel consumption may reduce considerably thanks to automated driving systems.

The control of vehicles can be represented as a spectrum consisting of five levels: (1) manual driving, (2) driver assistance, (3) partially automated driving, (4) highly automated driving, and (5) fully automated driving (*Gasser & Westhoff, 2012*). The introduction of driver assistance systems to the public took place in the 1990s with the release

Corresponding author
Pavlo Bazilinskyy,
p.bazilinskyy@tudelft.nl

of Adaptive Cruise Control (ACC), a system that automates the longitudinal motion of the vehicle (*Beiker, 2012*). Advancements in cameras, radars, lasers, and artificial intelligence have led to the creation of systems that make partially automated driving possible. Partially automated driving systems not only control the longitudinal motion of a vehicle, but also its lateral motion. Examples of such systems are BMW's Traffic Jam Assistant (*BMW, 2013*), Volvo's ACC with steer assist (*Volvo, 2013a*), and Mercedes' Distronic Plus with Steering Assist (*Daimler, 2013*). In partially automated driving, drivers are usually required to keep their eyes focused on the road and intermittently touch the steering wheel.

Highly automated driving (HAD) is a next step. In HAD, the human can release the hands from the steering wheel and is no longer required to monitor the road permanently (e.g., *Banks, Stanton & Harvey, 2014*). However, humans still have an important role in the control of highly automated vehicles (*Alicandri & Moyer, 1992*; *Dingus, Hulse & Barfield, 1998*; *Levitan, Golembiewski & Bloomfield, 1998*). In HAD, drivers can be asked to take over control of the vehicle when required, for example, when the vehicle automation cannot solve a task in a demanding traffic environment. The time between issuing a 'takeover request' and the required moment of transition of control from the vehicle to the human is a critical design parameter (*Gold et al., 2013*). If the driver spends too much time on reclaiming the control of the vehicle, or if the driver does not comprehend the warning signal sent by the vehicle, an accident may result. Clearly, the design of appropriate feedback is essential for the successful introduction of HAD to the public roads. Indeed, inappropriate feedback is regarded as a primary cause of automation-induced accidents (*Norman, 1990*).

Fully automated driving (FAD) will be the next and final iteration in automated driving. People have been envisioning this step in the development of transportation for a long time. Almost half a millennium ago, Leonardo Da Vinci envisioned a pre-programmed clockwork cart (*Weber, 2014*). In 1939 during the New York World's Fair, General Motors presented their vision of the world 20 years into the future (1959–1960). In their Futurama exhibition, they introduced a concept of automated highways with trench-like lanes for separating traffic (*Wetmore, 2003*). In 1953, the futurist Isaac Asimov wrote a short story 'Sally' that pictured a situation where only cars that did not require a human driver were allowed on the roads.

FAD offers numerous potential benefits. It could reduce stress and allow the operator to engage in non-driving tasks such as working, using in-vehicle entertainment, or resting (e.g., *Jamson et al., 2011*; *Llaneras, Salinger & Green, 2013*). Furthermore, FAD is a recommended solution for achieving an optimal traffic flow, for example by means of platooning on highways (*Bergenhem et al., 2012*; *Varaiya, 1993*). The Google Driverless Car is one of the existing prototypes of FAD (*Markoff, 2010*). However, this particular vehicle does not fully comply with the principles of FAD; in reality, the Google Driverless Car relies on accurate three-dimensional maps of the environment and currently cannot cope with all dynamic environments of high complexity. It requires considerable advances in sensing and artificial intelligence before FAD becomes practically feasible on all public roads. Continental, a leading German manufacturer specialising on components for automotive

industry, predicts that FAD will be launched in the year 2025 (*Continental, 2012*), whereas some voices have argued that FAD will never happen (*Gomes, 2014*; *Underwood, 2014*; *Yoshida, 2014*).

Although automated driving systems are expected to improve safety, certain side effects may occur regarding the human factor (e.g., *Bainbridge, 1983*; *Desmond, Hancock & Monette, 1998*; *Merat et al., 2012*; *Brandenburg & Skottke, 2014*). A degraded reaction time to critical events has been found among drivers exposed to ACC (*Stanton, Young & McCaulder, 1997*; *Stanton et al., 2001*; *Rudin-Brown & Parker, 2004*; *Larsson, Kircher & Hultgren, 2014*), and this issue is likely to be aggravated in higher levels of automated driving (*De Winter et al., 2014*; *Strand et al., 2014*). Furthermore, it is expected that people who will be driving highly and fully automated cars will suffer from a reduction of their manual control skills, similar to pilots in highly automated airplanes (*Ebbatson, 2009*; *Scallen, Hancock & Duley, 1995*). The development of effective feedback systems is considered important in supporting operator's sustained attention, also called vigilance (*Heikoop et al., submitted for publication*).

## Auditory displays

As mentioned above, unless the driving task is fully automated, an appropriate feedback system is required that warns and/or informs the human when automation mode changes are required. The present study investigated the potential of auditory feedback in automated driving. The auditory modality has several important characteristics: (1) it is omnidirectional. That is, unlike visual cues, auditory cues can be received from any direction. This is especially important in automated driving, during which the driver may not be attending to the road and dashboard; (2) the auditory sense can receive information at almost all times; (3) sound is transient, that is, unlike visual information which can be continuously available, information passed in the form of sound is only available at that particular moment; (4) although auditory cues may be masked by other sounds, humans have the ability to selectively focus on one sound when multiple streams of sound are available, also known as the cocktail party effect (*Bregman, 1990*; *Cooke & Ellis, 2001*; *Hermann, Hunt & Neuhoff, 2011*; *Wickens & Hollands, 2013*).

An advantage of sound is that it is possible to use language, which may be more informative as compared to the information conveyed with haptic or visual interfaces. Because of the aforementioned qualities of sound, auditory displays are used in a variety of applications, especially in those cases where the user needs to be alerted or where additional visual load has to be avoided. For example, the majority of present route navigation devices use voice and sound messages to give directions to their users (*Holland, Morse & Gedenryd, 2002*), and flight crews use auditory signals to get informed about proximate aircraft or to obtain directional information (e.g., *Begault, 1993*; *Bronkhorst, Veltman & Van Breda, 1996*). An auditory interface in combination with tactile feedback was suggested in a driving simulator study (*Ho, Reed & Spence, 2007*) as an optimal warning system for collision avoidance. The auditory modality has potential not only as a warning method, but also for providing input to the machine (e.g., speech interfaces).

Literature reviews (*Stanton & Edworthy, 1999*; *Barón & Green, 2006*) suggest that people drive 'better' (i.e., lower lane variation, steadier speed) when auditory interfaces are employed in a manually driven car.

Auditory feedback can be delivered as a pre-recorded voice or as an artificial sound warning/message. The term earcon refers to a brief auditory message (e.g., a tune or a sound of a bell) that represents a certain event or object. Earcons have been introduced to desktop computers to complement visual icons (*Mynatt, 1990*; *Belz, Robinson & Casali, 1999*; *Hermann, Hunt & Neuhoff, 2011*). Previous research has shown that a female voice is favoured over a male voice in route navigation devices (*Large & Burnett, 2013*). However, national or cultural differences seem to exist, where in some cases, the male voice is preferred over the female voice. In 2010, BMW supposedly had to recall its navigating system in Germany because male drivers disliked the idea of following orders communicated via a female voice (*Takayama & Nass, 2008*), and Apple recently added the option of a male voice to their voice control system Siri (*Bosker, 2013*). In a driving simulator study by *Jonsson & Dahlbäck (2011)*, non-native speakers of English responded more accurately to route instructions provided by a female voice than to route instructions provided by a male voice.

## Auditory systems in current vehicles: parking assistant and forward collision warning system

Modern vehicles often include systems that assist in driving and increase road safety. Such systems support drivers by providing auditory/visual/haptic warning messages and by taking over control of some of the driving tasks. In the present survey, we investigated the opinion of people on two existing auditory systems: a parking assistant (PA) and a forward collision warning system (FCWS).

The first generations of PAs were so-called parking sensors, which produce warning sounds (beeps) when the car gets too close to a nearby object while parking, using ultrasonic or electromagnetic sensors (*BMW, 2013*; *Toyota, 2014*; *Volkswagen, 2014*). Some recent PA systems take over the positioning of the vehicle during parking, leaving the control of acceleration and deceleration to the driver (*Volkswagen, 2014*). Other PAs take over control of the parking process entirely, as can be seen in the Toyota Prius 2015 and BMW X5 (*BMW, 2014*; *Toyota, 2014*).

A FCWS is a system that provides a warning sound when a vehicle is rapidly approaching a vehicle in front. FCWSs have the potential to prevent a large portion of rear-end collisions (*Jamson, Lai & Carsten, 2008*; *Kingsley, 2009*; *Kessler et al., 2012*). If a potential accident is detected by the FCWS, the system either gives a warning to the driver (*Honda, 2014*) or engages in emergency braking and/or steers way from the object (*Volvo, 2013b*). Most FCWS detect vehicles with the help of computer vision (*Srinivasa, 2002*; *Dagan et al., 2004*), an approach that is used by companies like Honda and BMW (*BMW, 2013*; *Volvo, 2013b*; *Honda, 2014*) and/or radars (*Volvo, 2013b*; *Ford, 2014*; *Honda, 2014*; *Mercedes-Benz, 2014*). Both approaches have limitations, and the system may not issue warnings or stop the vehicle in bad weather or in other situations where the sensors are obscured by external

factors. The introduction of vehicle-to-vehicle (V2V) communication may increase the efficiency and capabilities of collision warning systems (e.g., *Miller & Huang, 2002*). Eighty-eight percent of owners of Volvo cars surveyed by *Braitman et al. (2010)* reported always having the FCWS turned on.

It is expected that both PAs and FCWSs will remain in future partially and highly automated vehicles. However, these technologies will become obsolete with the introduction of FAD because both parking and collision avoidance will be handled without any input from the human driver.

### 'Augmented/spatial' sound system for fully automated driving

Auditory warning signals will not be required in FAD, because in FAD the automation by definition takes care of all possible emergency conditions. This study proposes an experimental setup aimed at the three-dimensional augmentation of sound surrounding a vehicle, hereafter referred to as the 'future system' (FS), which could be used in FAD for entertainment and comfort. Three-dimensional sound is being developed as a means for providing feedback to humans (*Lumbreras, Sánchez & Barcia, 1996*; *Garas, 2000*; *Rozier, 2000*; *Godinho, António & Tadeu, 2001*; *Dobler, Haller & Stampfl, 2002*).

Our proposed FS filters out unwanted sounds (e.g., tire/engine noise coming from vehicles in the vicinity) and amplifies desired sounds (e.g., sound of birds singing in a park). We envision that such a system could be used in future fully automated vehicles. Vehicles driving fully automatically have full control of the vehicle and must have reliable detection capabilities of the environment. Drivers of such vehicles will not be required to pay attention to the processes that take place in the environment surrounding the car. Hence, a spatial augmentation of sounds that a driver prefers to hear and simultaneous cancelation of unwanted sounds may enhance the pleasure of being engaged in FAD. Such system will probably have to be configurable: drivers must have the option to select which sounds they want to augment and which sounds they wish to filter out, as well as to adjust the volume of these sounds.

### The aim of the present survey study

As mentioned above, feedback is important in HAD, especially regarding transitions of control. It is relevant for the development of automated driving systems to know what types of interfaces people want and need. Because automated cars do not exist yet on the consumer market, it is impossible to test such research questions in an ecologically valid environment, except through driving simulator research.

The present study was undertaken from a different point of view. We proceeded on the basis that respondents were asked to *imagine* automated driving scenarios. The aim of the present study was to investigate the opinion of people on two existing auditory displays (PA & FCWS) as well as the augmented sound system 'FS'. The respondents were asked to judge two qualities of the systems—helpfulness and annoyance—and state whether they would consider using automated versions of such systems in the future. In addition, we asked people to report their preferred type of feedback for takeover requests in HAD. Statistical associations between self-reported driving style as measured with the Driver Behaviour

Questionnaire (DBQ), yearly mileage, number of accidents, and opinions of respondents on the qualities of the proposed systems were assessed.

The hypothesis that people from non-English speaking countries prefer a female voice to a male voice in automated driving systems was also tested. Additionally, the respondents were asked to provide their general thoughts on the concept of automated driving in a free-response question. Finally, the respondents provided their opinion on the year of introduction of fully automated driving in their country of residence. Results of these analyses were compared with findings from two previous surveys that asked questions related to other aspects of automated vehicles (*De Winter et al., 2015*; *Kyriakidis, Happee & De Winter, 2015*).

## METHODS

### Survey

A survey containing 31 questions was developed with the online tool CrowdFlower (www.crowdflower.com). Table 1 shows the questions of the survey as well as the corresponding coding. The full survey is included in the Supplemental Information 1. The survey was targeted towards reasonably educated persons without knowledge of automated driving. A previous survey indicated that people who work on CrowdFlower-based surveys have mostly undergraduate degrees (*Kyriakidis, Happee & De Winter, 2015*).

The present survey introduced in plain language three levels of driving: manual driving, partially automated driving, and fully automated driving. Manual driving was referred to as "normal (non automated) cars". The explanation of partially driving was provided as follows: "Imagine again that you are driving in an automated car (that can perform certain tasks without any interaction from the humans in the car). However, the automation cannot handle all possible situations, and you sometimes have to take over control". Respondents were asked to imagine fully automated driving as follows: "Imagine a fully automated car (no steering wheel) that drives completely on its own with no manual interaction".

The survey contained questions on the person's age, gender, driving frequency, mileage, and accident involvement. The questions asking participants to provide information on their driving style were based on the violations scale of the DBQ, as used by *De Winter (2013)*.

The respondents were asked to express their opinion on two currently existing systems and one proposed setup that could be used during fully automated driving. Specifically, we asked respondents about (1) a parking assistant (PA) in a manually driven car that produces warning sounds (beeps) when the car gets too close to a nearby object while parking, (2) a forward collision warning system (FCWS) in a manually driven car that provides a warning sound when a car is rapidly approaching another car in front, and (3) a future augmented surround sound system in a fully automated vehicle (FS). The FS was described as follows: "Now imagine that this fully automated car records what is happening outside and plays it via speakers inside the car, informing the occupants about the outside environment. In other words, those who sit in the car can hear what is happening outside

**Table 1** All survey items.

| Variable | Question | Full question as reported in the survey | Used coding |
|---|---|---|---|
| Instr | Q1 | Have you read and understood the above instructions? | 1 = Yes, 2 = No |
| Gender | Q2 | What is your gender? (1 = female, 2 = male) | −1 = I prefer not to respond, 1 = Female, 2 = Male |
| Age | Q3 | What is your age? | Positive integer value |
| DriveFreq | Q4 | On average, how often did you drive a vehicle in the last 12 months? | −1 = I prefer not to respond, 1 = Never, 2 = Less than once a month, 3 = Once a month to once a week, 4 = 1 to 3 days a week, 5 = 4 to 6 days a week, 6 = Every day |
| KmYear | Q5 | About how many kilometres (miles) did you drive in the last 12 months? | −1 = I prefer not to respond, 1 = 0, 2 = 1–1,000, 3 = 1,001–5,000, 4 = 5,001–15,000, 5 = 15,001–20,000, 6 = 20,001–25,000, 7 = 25,001–35,000, 8 = 35,001–50,000, 9 = 50,001–100,000, 10 = more than 100,000 |
| NrAcc | Q6 | How many accidents were you involved in when driving a car in the last 3 years? (please include all accidents, regardless of how they were caused, how slight they were, or where they happened)? | −1 = I prefer not to respond, 1 = 0, 2 = 1, 3 = 2, 4 = 3, 5 = 4, 6 = 5, 7 = More than 5 |
| Vangered | Q7 | How often do you do the following?: Becoming angered by a particular type of driver, and indicate your hostility by whatever means you can. | −1 = I prefer not to respond, 1 = 0 times per month, 2 = 1 to 3 times per month, 3 = 4 to 6 times per month, 4 = 7 to 9 times per month, 5 = 10 or more times per month |
| Vmotorway | Q8 | How often do you do the following? Disregarding the speed limit on a motorway. | −1 = I prefer not to respond, 1 = 0 times per month, 2 = 1 to 3 times per month, 3 = 4 to 6 times per month, 4 = 7 to 9 times per month, 5 = 10 or more times per month |
| Vresident | Q9 | How often do you do the following? Disregarding the speed limit on a residential road. | −1 = I prefer not to respond, 1 = 0 times per month, 2 = 1 to 3 times per month, 3 = 4 to 6 times per month, 4 = 7 to 9 times per month, 5 = 10 or more times per month |
| Vfollowing | Q10 | How often do you do the following? Driving so close to the car in front that it would be difficult to stop in an emergency. | −1 = I prefer not to respond, 1 = 0 times per month, 2 = 1 to 3 times per month, 3 = 4 to 6 times per month, 4 = 7 to 9 times per month, 5 = 10 or more times per month |
| Vrace | Q11 | How often do you do the following? Racing away from traffic lights with the intention of beating the driver next to you. | −1 = I prefer not to respond, 1 = 0 times per month, 2 = 1 to 3 times per month, 3 = 4 to 6 times per month, 4 = 7 to 9 times per month, 5 = 10 or more times per month |

Table 1 (*continued*)

| Variable | Question | Full question as reported in the survey | Used coding |
|---|---|---|---|
| Vhorn | Q12 | How often do you do the following? Sounding your horn to indicate your annoyance with another road user. | −1 = I prefer not to respond, 1 = 0 times per month, 2 = 1 to 3 times per month, 3 = 4 to 6 times per month, 4 = 7 to 9 times per month, 5 = 10 or more times per month |
| Vphone | Q13 | How often do you do the following? Using a mobile phone without a hands free kit. | −1 = I prefer not to respond, 1 = 0 times per month, 2 = 1 to 3 times per month, 3 = 4 to 6 times per month, 4 = 7 to 9 times per month, 5 = 10 or more times per month |
| Vmean | N/A | Mean for Q7–12 | Numeric value |
| PApast | Q14 | In the past month, did you drive a car with a parking assistant? | −1 = I prefer not to respond, 1 = I do not know, 2 = No, 3 = Yes |
| PAhelp | Q15 | A parking assistant is helpful. | −1 = I prefer not to respond, 1 = Disagree strongly, 2 = Disagree a little, 3 = Neither agree nor disagree, 4 = Agree a little, 5 = Agree strongly |
| PAannoy | Q16 | A parking assistant is annoying. | −1 = I prefer not to respond, 1 = Disagree strongly, 2 = Disagree a little, 3 = Neither agree nor disagree, 4 = Agree a little, 5 = Agree strongly |
| PAopin | Q17 | What do you think are the disadvantages of a parking assistant? | Textual response |
| PAfut | Q18 | I would like to have a system in my car that can park the car automatically, just by pressing a button. | −1 = I prefer not to respond, 1 = Disagree strongly, 2 = Disagree a little, 3 = Neither agree nor disagree, 4 = Agree a little, 5 = Agree strongly |
| FCWSpast | Q19 | In the past month, did you drive a car with a forward collision warning system? | −1 = I prefer not to respond, 1 = I do not know, 2 = No, 3 = Yes |
| FCWShelp | Q20 | A forward collision warning system is helpful. | −1 = I prefer not to respond, 1 = Disagree strongly, 2 = Disagree a little, 3 = Neither agree nor disagree, 4 = Agree a little, 5 = Agree strongly |
| FCWSannoy | Q21 | A forward collision warning system is annoying. | −1 = I prefer not to respond, 1 = Disagree strongly, 2 = Disagree a little, 3 = Neither agree nor disagree, 4 = Agree a little, 5 = Agree strongly |
| FCWSfut | Q22 | I would you like to have a system in my car that brakes automatically to avoid collisions (Autonomous Emergency Braking). | −1 = I prefer not to respond, 1 = Disagree strongly, 2 = Disagree a little, 3 = Neither agree nor disagree, 4 = Agree a little, 5 = Agree strongly |
| FCWSopin | Q23 | What do you think are the disadvantages of a forward collision warning system? | Textual response |
| FSannoy | Q24 | I believe that this type of surround sound system would be annoying. | −1 = I prefer not to respond, 1 = Disagree strongly, 2 = Disagree a little, 3 = Neither agree nor disagree, 4 = Agree a little, 5 = Agree strongly |

Table 1 (*continued*)

| Variable | Question | Full question as reported in the survey | Used coding |
|---|---|---|---|
| FSfut | Q25 | I would prefer to use such a sound system instead of opening the window, when driving through a scenic place (for example, a national park). | −1 = I prefer not to respond, 1 = Disagree strongly, 2 = Disagree a little, 3 = Neither agree nor disagree, 4 = Agree a little, 5 = Agree strongly |
| FSopin | Q26 | What would be the advantages and the disadvantages of such sound system? | Textual response |
| TORint | Q27 | Now imagine again that you are driving in an automated car (that can perform certain tasks without any interaction from the humans in the car). However, the automation cannot handle all possible situations, and you sometimes have to take over control. What type of warning signal would you like to receive in case manual take over is required? | 1 = Warning sound: one beep, 2 = Warning sound: two beeps, 3 = Warning sound: horn sound, 4 = Warning sound: bell sound, 5 = Warning light, 6 = Visual warning message projected on windscreen 'Take over please', 7 = Vibrations in your seat, 8 = Vibrations in your steering wheel, 9 = Vibrations in your seatbelt, 10 = Vibrations in the floor, 11 = Female voice: 'Take over please', 12 = Male voice: 'Take over please', 13 = Other, 14 = None of the above |
| TORintot | Q28 | If you answered 'Other' in the previous question, please specify what type of warning signal you would like to receive in the described scenario. | Textual response |
| FACpref | Q29 | I would prefer to drive in a fully automated car rather than a normal (non automated) car. | −1 = I prefer not to respond, 1 = Disagree strongly, 2 = Disagree a little, 3 = Neither agree nor disagree, 4 = Agree a little, 5 = Agree strongly |
| YearAuto | Q30 | In which year do you think that most cars will be able to drive fully automatically in your country of residence? | Year |
| Comm | Q31 | Please provide any suggestions which could help engineers to build safe and enjoyable automated cars. | Textual response |
| SurvTime | | Survey time (*derived from results generated by CrowdFlower*) | Seconds |

even when their windows are closed. Sound volume in such system could be adjusted; particular noise (for example sound coming from another vehicle) could be filtered out. Such a system could, for example, be used during a leisure drive through a park on a hot day". Illustrations belonging to the three scenarios (i.e., PA, FCWS, FS) were used in the survey (Fig. 1). No auditory examples were used. The illustrations were uploaded to a remote site in order to be embedded to the survey. Supplemental Information 1 contains the XML code used to create the survey. If one wishes to add images to a CrowdFlower survey, the suggested method could be used.

The respondents were asked to indicate disadvantages of the PA (Q17) and FCWS (Q23) and to indicate advantages and disadvantages of the FS (Q26) by means of textual

**Figure 1** **Illustrations belonging to the three scenarios presented to the respondents.** (A) Parking assistant (PA); (B) Forward collision warning system (FCWS); (C) Future system (FS).

responses. The respondents also had the opportunity to indicate the preferred mode of feedback for receiving a takeover request (Q27 & Q28). In the last question (Q31), they were asked to "provide any suggestions, which could help engineers to build safe and enjoyable automated cars". Giving a response to this last free-response question was optional. All examples of given comments shown in this article are direct quotes from the responses; no grammatical or syntactic errors were corrected. The respondents had to complete all questions (except Q28 & Q31), and each question had an *I prefer not to respond* response option.

## Configuration of CrowdFlower

In the instructions, the respondents were informed that they would need approximately 10 min to complete the survey. The task expiration time was set to 30 min. Contributors from all countries were allowed to participate in the survey, in order to collect data from an as large and diverse population as possible. Moreover, the lowest level of experience of contributors, 'Level 1 contributors', was selected. This level of experience accounts for 60% of completed work on CrowdFlower. As a result, the survey was available to a large number of workers, which allowed reaching a relatively diverse group of users of the platform. Completing the survey more than once from the same IP address was allowed (note, however, that multiple responses from the same IP address were filtered out in our analyses, see results section). For the completion of the survey a payment of $0.15 was offered, and 2,000 responses were collected. The study was preceded by a pilot test with 10 respondents. The pilot test did not lead to any changes in the survey, and these 10 respondents were not included in the analysis.

## Analyses

Descriptive statistics (i.e., mean, median, standard deviation, skewness, and number of responses) were calculated for each of the variables. The skewness was calculated as the third central moment divided by the cube of the standard deviation. A Spearman correlation matrix among the variables was created. The first author manually performed the analysis of textual responses (Q17, Q23, Q26, Q28, & Q31).

CrowdFlower automatically provides the respondent's country based on his/her IP address. We analysed the preferences of people from English speaking countries, as defined by the UK government (*UK Visas & Immigration, 2014*: Antigua and Barbuda, Australia, Bahamas, Barbados, Belize, Canada, Dominica, Grenada, Ireland, Jamaica, New Zealand, Saint Lucia, Trinidad and Tobago, United Kingdom, and the United States) versus

non-English speaking countries regarding the use of a male or female voice for supporting takeover requests during highly automated driving. Supplemental Information 1 contains the MATLAB script used to analyse the data.

### Ethics statement

All data were collected anonymously. The research was approved by the Human Research Ethics Committee (HREC) of the Delft University of Technology. Documented informed consent was obtained via a dedicated survey item asking whether the respondent had read and understood the survey instructions.

## RESULTS

### Number of respondents and respondent satisfaction

In total, 2,000 surveys were completed. The responses were gathered on 2 September 2014 between 15:00 and 20:15 (CET). The survey received an overall satisfaction rating of 4.4 out of 5.0. Additionally, the respondents ranked clearness of the instructions as 4.4/5.0, fairness of the questions as 4.2/5.0, easiness of the survey as 4.2/5.0, and the offered payment as 4.1/5.0.

### Data filtering

The respondents who indicated they had not read the instructions ($N = 10$), who indicated they were under 18 and thereby did not adhere to the survey instructions ($N = 6$), who chose the *I prefer not to respond* or *I do not know* options in one or more of the multiple choice questions ($N = 231$), who indicated they never drive ($N = 193$), or who indicated they drive 0 km per year ($N = 191$) were excluded from the analyses. Since no limitations were applied on the number of responses that could be generated per IP address, some people completed the survey more than once. Such behaviour was seen as an indication that these persons participated in the survey primarily because of monetary gain. Thus, we applied a strict filter, and all data generated from non-unique IP addresses were removed ($N = 465$). In total, 795 surveys were removed, leaving 1,205 completed surveys for further analysis.

For the question "In which year do you think that most cars will be able to drive fully automatically in your country of residence?", non-numeric responses (e.g., a year complemented by words such as "maybe 2030", or "never") and answers before the year 2014 were excluded, leaving 1,082 numeric responses.

### Analyses at the individual level

The 1,205 respondents were from 91 countries (all 2,000 responses were associated with 95 countries). Descriptive statistics for all variables are listed in Table 2. The respondents took on average 9.2 min to complete the survey ($SD = 5.6$ min, median $= 7.7$ min). The Supplemental Information 1 contains the entire Spearman correlation matrix. The correlation coefficients between variables that related to questions about the PA, FCWS, and FS (PApast, PAhelp, PAannoy, PAfut, FCWSpast, FCWShelp, FCWSannoy, FCWSfut, FSannoy, & FSfut) on the one hand, and Age, DriveFreq, KmYear, NrAcc, the

**Table 2 Descriptive statistics for the survey items (N = 1,205).** The response option "I don't know" was omitted for the variables PApast and FCWSpast.

| Variable | Mean | Median | SD | Skewness | Min | Max |
|---|---|---|---|---|---|---|
| Gender | 1.75 | 2 | 0.43 | −1.17 | 1 | 2 |
| Age | 31.94 | 30 | 10.49 | 1.04 | 18 | 73 |
| DriveFreq | 4.72 | 5 | 1.21 | −0.66 | 2 | 6 |
| KmYear | 4.09 | 4 | 1.78 | 0.92 | 2 | 10 |
| NrAcc | 1.47 | 1 | 0.94 | 2.88 | 1 | 7 |
| Vangered | 1.86 | 2 | 0.86 | 1.46 | 1 | 5 |
| Vmotorway | 1.85 | 2 | 1.05 | 1.54 | 1 | 5 |
| Vresident | 1.70 | 1 | 1.01 | 1.79 | 1 | 5 |
| Vfollowing | 1.45 | 1 | 0.77 | 2.07 | 1 | 5 |
| Vrace | 1.32 | 1 | 0.69 | 2.62 | 1 | 5 |
| Vhorn | 1.86 | 2 | 1 | 1.41 | 1 | 5 |
| Vphone | 1.64 | 1 | 1.01 | 1.84 | 1 | 5 |
| Vmean | 1.67 | 1.57 | 0.57 | 1.36 | 1 | 4.71 |
| PApast | 2.27 | 2 | 0.45 | 1.03 | 2 | 3 |
| PAhelp | 4.33 | 5 | 0.88 | −1.38 | 1 | 5 |
| PAannoy | 2.35 | 2 | 1.18 | 0.39 | 1 | 5 |
| PAfut | 3.87 | 4 | 1.24 | −0.93 | 1 | 5 |
| FCWSpast | 2.10 | 2 | 0.30 | 2.65 | 2 | 3 |
| FCWShelp | 4.11 | 4 | 1.04 | −1.14 | 1 | 5 |
| FCWSannoy | 2.56 | 3 | 1.26 | 0.27 | 1 | 5 |
| FCWSfut | 3.77 | 4 | 1.22 | −0.80 | 1 | 5 |
| FSannoy | 3.21 | 3 | 1.22 | −0.18 | 1 | 5 |
| FSfut | 3.00 | 3 | 1.29 | −0.09 | 1 | 5 |
| FACpref | 3.01 | 3 | 1.33 | −0.05 | 1 | 5 |
| YearAuto | 2,078.33 | 2,030 | 713.77 | 30.73 | 2,014 | 25,000 |
| SurvTime | 553.95 | 462 | 338.41 | 1.34 | 58 | 1,810 |

DBQ variables (Vangered, Vmotorway, Vresident, Vfollowing, Vrace, Vhorn, & Vphone), YearAuto, and SurvTime, on the other, were overall small, between −0.15 and 0.25.

The respondents' mean and median age were 31.9 and 30 years, respectively. Figure 2 shows the distribution of the respondents in 5-year wide age groups. 75.2% of the respondents were male (906 men vs. 299 women). The frequencies of the answers are provided in Table 3.

Figure 3 shows that the respondents expected most cars to be able to drive in fully automated mode in their countries of residence around 2,030 (median response), with a highly skewed distribution.

The respondents were asked to provide their opinion on two characteristics of the PA and FCWS systems, annoyance and helpfulness, and whether they would be willing to have automated versions of such systems in their own cars (Q18 for the PA & Q22 for the FCWS), all questions on a scale from 1 (*disagree strongly*) to 5 (*agree strongly*). Figure 4 shows the results for these questions.

**Table 3 Frequencies of answers.**

| Variable | 1 | 2 | 3 | 4 | 5 | 6 | 7 | 8 | 9 | 10 |
|---|---|---|---|---|---|---|---|---|---|---|
| Gender | 299 | 906 | | | | | | | | |
| DriveFreq | 0 | 79 | 108 | 293 | 313 | 412 | | | | |
| KmYear | 0 | 250 | 245 | 337 | 148 | 76 | 77 | 49 | 16 | 7 |
| NrAcc | 855 | 226 | 79 | 19 | 13 | 6 | 7 | | | |
| Vangered | 417 | 629 | 96 | 33 | 30 | | | | | |
| Vmotorway | 541 | 466 | 97 | 39 | 62 | | | | | |
| Vresident | 663 | 379 | 78 | 32 | 53 | | | | | |
| Vfollowing | 807 | 295 | 70 | 20 | 13 | | | | | |
| Vrace | 939 | 182 | 60 | 15 | 9 | | | | | |
| Vhorn | 513 | 480 | 123 | 44 | 45 | | | | | |
| Vphone | 730 | 304 | 89 | 34 | 48 | | | | | |
| PApast | 17 | 865 | 323 | | | | | | | |
| PAhelp | 14 | 39 | 134 | 367 | 651 | | | | | |
| PAannoy | 383 | 297 | 290 | 196 | 39 | | | | | |
| PAfut | 86 | 109 | 175 | 338 | 497 | | | | | |
| FCWSpast | 28 | 1,058 | 119 | | | | | | | |
| FCWShelp | 34 | 68 | 178 | 373 | 552 | | | | | |
| FCWSannoy | 331 | 254 | 324 | 208 | 88 | | | | | |
| FCWSfut | 82 | 123 | 203 | 377 | 420 | | | | | |
| FSannoy | 126 | 218 | 346 | 309 | 206 | | | | | |
| FSfut | 204 | 222 | 312 | 301 | 166 | | | | | |
| FACpref | 204 | 252 | 264 | 292 | 193 | | | | | |

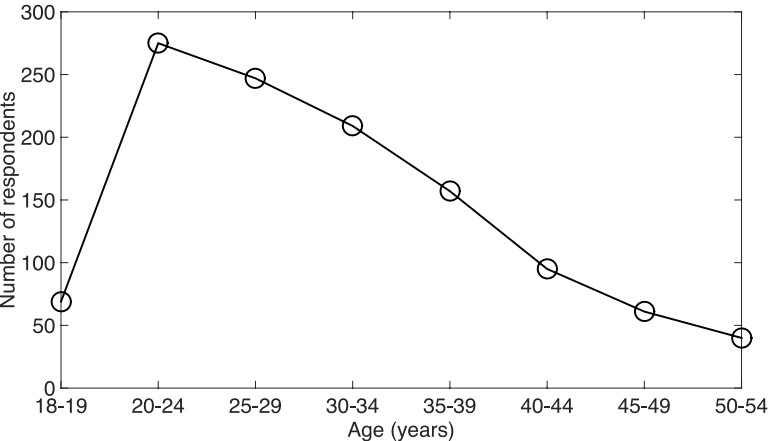

**Figure 2 Distribution of the age of the respondents aged between 18 and 54 years.**

Figure 5 shows associations between the opinion of the respondents on annoyance and helpfulness of the PA and FCWS and their age divided into 5-year wide bins. Figure 5A shows that younger respondents found that both the PA ($\rho = -0.05, p = .103$) and the FCWS ($\rho = -0.14, p < .001$) were more annoying, but these effects were weak. The

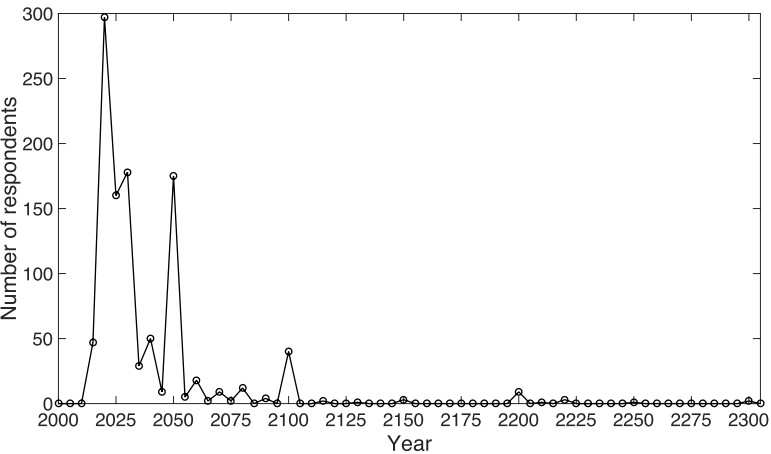

**Figure 3** Distribution of responses for the question: "*In which year do you think that most cars will be able to drive fully automatically in your country of residence?*" **(Q30).** Years were divided into 5-year-wide bins.

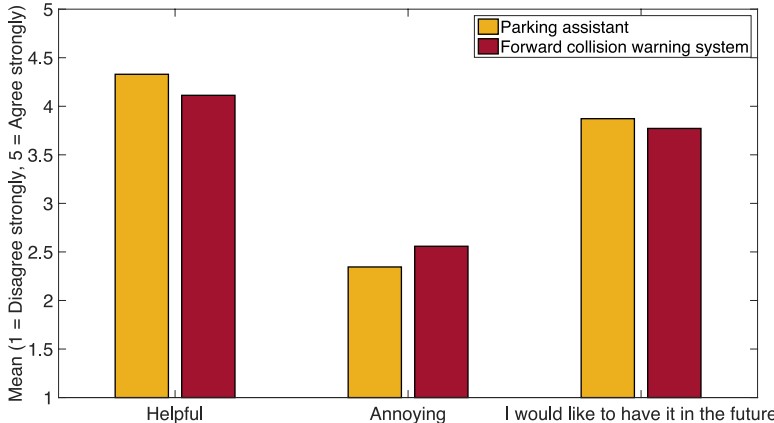

**Figure 4** Opinion of the respondents on whether a PA and FCWS are helpful and annoying, and whether they would like to have automated versions of such systems in their cars in the future.

Spearman correlation between the respondents' age and the reported annoyance of the FS was weak as well ($\rho = 0.06, p = .035$). Figure 5B shows that the perceived helpfulness of the FCWS ($\rho = 0.12, p < .001$) slightly increased with age. People who found the PA annoying typically indicated that the FCWS was also annoying ($\rho = 0.47, p < .001$), and respondents who thought that the PA was helpful, considered the FCWS to be helpful as well ($\rho = 0.34, p < .001$).

Figure 6 shows the respondents' opinion on the proposed future system. The respondents were asked whether they would find such a system annoying (Q24) and whether they would prefer to use such a system instead of opening windows while driving in a fully automated car through a scenic place (Q25). A large portion of the respondents was neutral in their responses: 346 people chose the option *Neither agree nor disagree* in Q24, and 312 persons chose the same option in Q25.

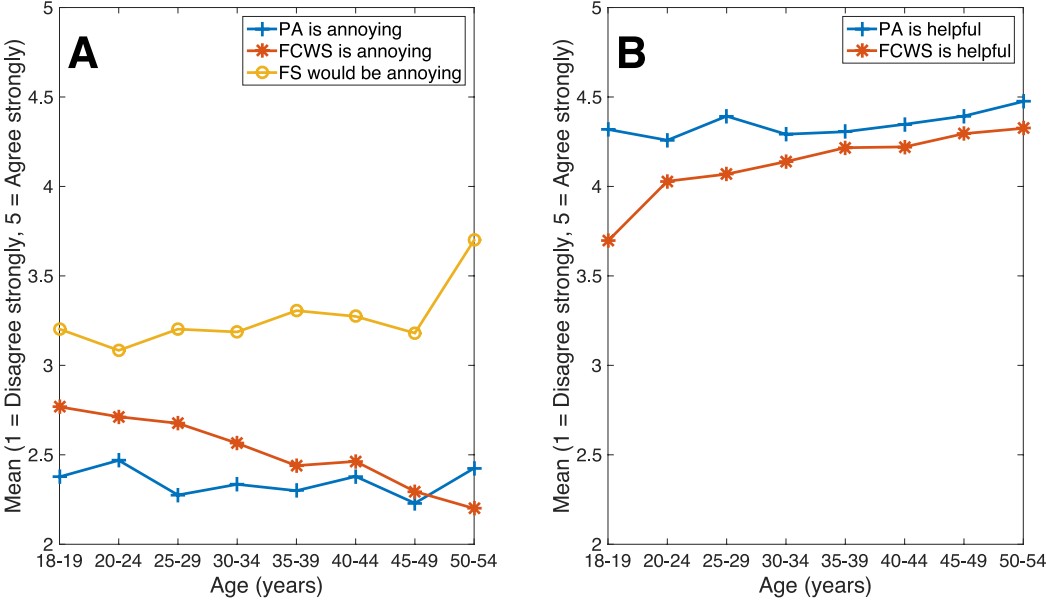

**Figure 5** Opinion on the annoyance and helpfulness of the parking assistant (PA), forward collision warning system (FCWS), and future system (FS). (A) Opinion on the annoyance of the PA (Q16), FCWS (Q21), and FS (Q24) as a function of age; (B) Opinion on the helpfulness of the PA (Q15) and FCWS (Q20) as a function of age. Age was divided into 5-year-wide bins.

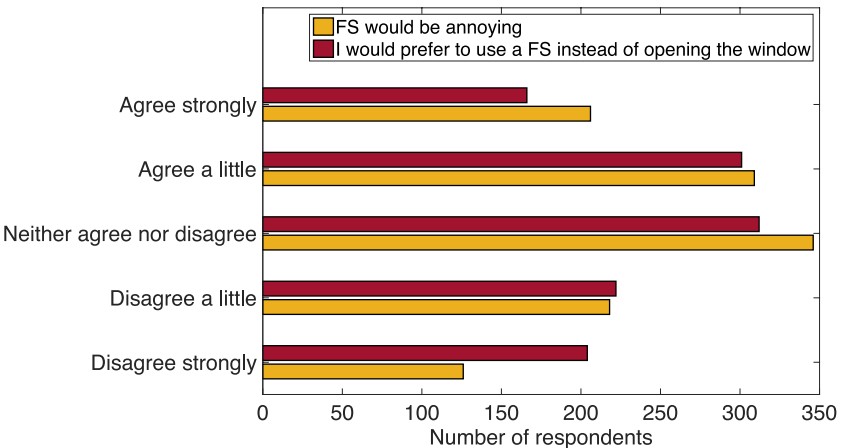

**Figure 6** Distribution of opinions on whether the proposed future system (FS) would be annoying (Q24) and whether the respondents would prefer the system to opening windows in fully automated cars (Q25).

In Q27 the respondents were asked to report on the types of feedback that they would like to be supported by in case of a takeover request during highly automated driving. The respondents were allowed to select multiple options. Figure 7 shows that a large number of people preferred auditory feedback provided by a female voice saying 'Take over please' ($N = 514$). The number of respondents who chose the option with the male voice was considerably lower ($N = 244$). Figure 7 makes a distinction between the numbers of female

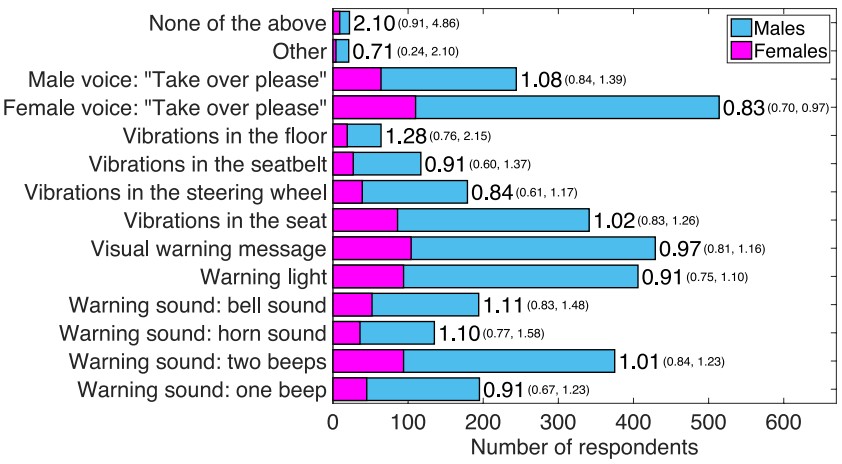

**Figure 7 Preference for particular takeover requests during highly automated driving.** Numbers of respondents who indicated a preference for a particular takeover request during highly automated driving in the question: "*Now imagine again that you are driving in an automated car (that can perform certain tasks without any interaction from the humans in the car). However, the automation cannot handle all possible situations, and you sometimes have to take over control. What type of warning signal would you like to receive in case manual take over is required?*" (Q27). Each bar is supplemented by the corresponding 'risk ratios' of female respondents, calculated as the proportion of females who indicated this answer divided by the proportion of males who indicated this answer. If the risk ratio is greater than 1, females are overrepresented. Conversely, if the risk ratio is smaller than 1, females are underrepresented. 95% confidence intervals are shown in parentheses.

and male respondents. It is apparent that both female and male respondents preferred the female over the male voice.

Other types of auditory feedback were reported in the following order: two beeps ($N = 375$), one beep ($N = 195$), a bell sound ($N = 194$), and a horn sound ($N = 135$). The respondents indicated a high level of support for both visual signals offered in the question: a warning message projected on the windscreen 'Take over please' ($N = 429$) and a warning light ($N = 406$). However, respondents showed a relatively low level of acceptance of the offered variations of a vibration interface: vibrations in the seat ($N = 341$), vibrations in the steering wheel ($N = 179$), vibrations in the seatbelt ($N = 117$), and vibrations in the floor ($N = 64$). Furthermore, the results seem to suggest that female respondents were less likely than male respondents to prefer a female voice.

Figure 8 shows the opinion of the respondents on the combinations of warning signals. The figure shows that most people ($N = 188$) preferred a sound signal (i.e., one or two beeps, a horn, or bell) without additional information. A large number of people indicated that they would like to receive a combination of all four modalities ($N = 170$) or the combination of a sound signal, a visual message, and a voice ($N = 101$).

## Cross-national differences in opinion for feedback for takeover requests

Next, we tested the hypothesis whether peoples' preference for a female and male voice in supporting takeover requests in highly automated driving was different between English and non-English speaking countries. Figure 9 presents a scatter plot, showing the numbers

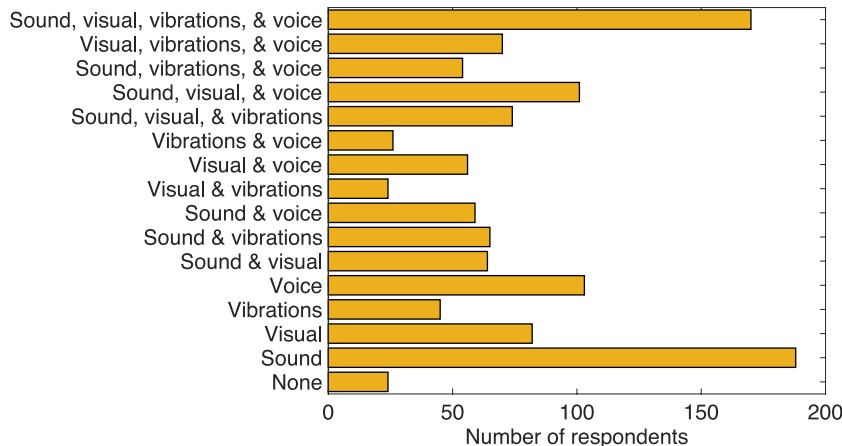

**Figure 8** **Preference to combinations of types of signals for aiding takeover requests during highly automated driving (Q27).** All possible combinations are listed. Hence, the total number of respondents adds up to 1,205.

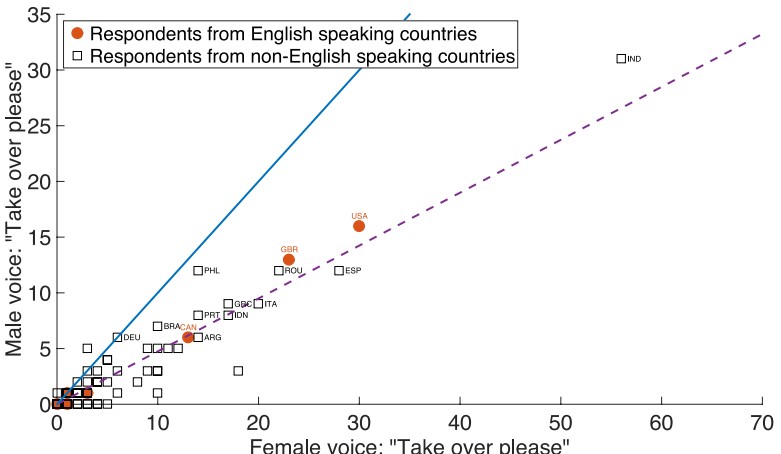

**Figure 9** **Numbers of respondents from English and non-English speaking countries who indicated a preference for a male and a female voice for a takeover request during highly automated driving (Q27).** The dashed line represents the ratio between the number of respondents who preferred a female voice and the number of respondents who preferred a male voice. The solid line is the line of unity. No labels are shown for countries with five or less respondents indicating a male voice, to support the clarity of the figure. Country abbreviations are listed according to ISO 3166-1 alpha-3.

of respondents per country who indicated that they would like to receive a female or a male voice. The overall percentage of respondents who expressed preference for a female voice was 43% (514/1,205), and the overall percentage of people who expressed preference for a male voice was 20% (244/1,205). The corresponding percentages were 42% (71/168) and 22% (37/168) for English speaking countries, and they were 43% (443/1,037) and 20% (207/1,037) for non-English speaking countries. The differences between English speaking countries and non-English speaking countries were not statistically significant (female voice: RR = 0.99, 95% CI [0.82–1.20]; male voice: RR = 1.10, 95% CI [0.81–1.50]).

## Analyses of textual comments

The respondents provided their feedback on the disadvantages of the PA in Q17. The responses that were less than five characters long ($N = 181$) or that were not written in English ($N = 39$) were ignored. Comments were processed before data filtering and were hence based on all 2,000 responses. 12.4% of the respondents ($N = 151$) provided negative feedback on the auditory interfaces in parking assistants. Many people ($N = 135$) indicated that PA systems were annoying, for example: "*Sound should not be too loud and annoying*" and "*I think it could be annoying especially when your focusing*". Thirty-seven respondents pointed out that the PA used overly loud sounds. Several answers to the question contained comments that the PA sounds can be distracting ($N = 21$) and inaccurate ($N = 48$). Five respondents indicated that they would prefer feedback in other types of modalities, for example: "*annoying, use something else instead of the constant loud beeping sounds*" and "*The sound, a voice message would be better*". Five respondents indicated that the PA systems cannot be used by deaf people.

The respondents indicated their opinion on the disadvantages of the FCWS in Q23. The responses that were less than five characters long ($N = 276$) or that were not written in English ($N = 35$) were not included in the analysis. Sixteen respondents indicated that the auditory feedback used in FCWS was annoying, for example: "*This situation might come up too often so the warning sound may get annoying fast*" and "*The beeps might feel annoying*".

Next, the respondents were asked to comment on possible advantages and disadvantages of the FS in Q26. The responses that were less than five characters long ($N = 138$) or that were not written in English ($N = 46$) were not included in the analysis. In total, 1,249 comments were analysed. A collection of mixed responses was received. Overall, more comments were classified as positive ($N = 132$) than negative ($N = 52$) to the FS. However, the respondents also pointed out concerns about a number of characteristics that they associated with the system: annoyance to both the driver and to other road users in the traffic ($N = 101$), distraction ($N = 47$), and loudness ($N = 28$). Fifty-five respondents expressed their concerns that the system would be impractical; however, most of such concerns could be associated with the lack of understanding of the concept of a fully automated car. Certain respondents showed a high level of negativity caused by an apparent lack of understanding the concept of filtering only specific sounds coming from the outside environment. Examples are: "*You can not hear some bells or signal from other cars*", "*Main disadvantage: makes driver unaware of any dangers*", "*If car noises are filtered out how would you hear if another car is incoming*", and "*I feel that filtering other car noise may be dangerous*".

In Q27 the respondents were asked to indicate their preference for types of interfaces to be used for takeover requests in HAD. One of the options in that question was "*Other*". If respondents selected this option, they were invited to provide further comments in Q28. The responses that were less than five characters long ($N = 32$) or that were not written in English ($N = 1$) were ignored. In total, 22 responses were analysed. One respondent indicated that he/she would prefer to be aided by continuous beeps until he/she reclaimed control. Another respondent stated "*steering wheel up or down motion to signal steering wheel usage needed, accompanied by a specific message*". One respondent mentioned that inter-

faces used in such scenario need to be adaptive depending on the urgency of the request *"It honestly depends on the situation the car needs me to take over for. Does it affect anyone's safety at all? Does it actually /need/ to be done straight away? Is it critically important in any other way? In those cases I'd obviously like a very noticeable signal however 'annoying' it may be. In other situations however I'd prefer a decent text message or a gentle reminder".*

## DISCUSSION

The aim of this study was to obtain opinions on preferred feedback types for takeover requests in HAD from a large number of people coming from all over the globe. Additionally, the aim was to measure perceived helpfulness and annoyance of auditory interfaces for three systems. Specifically, the respondents who participated in the survey were presented with two existing systems used in modern vehicles (a parking assistant [PA] & a forward collision warning system [FCWS]) and one futuristic setup (FS) envisioned for FAD. Respondents were asked whether they would consider using the proposed FS in future automated vehicles. Our survey helped us to gather opinions from people before technology is actually available.

Previous research suggests that the modality of aiding systems in automated cars should be chosen carefully to avoid frustration of people who will be using such vehicles and to increase safety of automation on public roads. *Stanton, Young & McCaulder (1997)* expressed concerns that interfaces currently employed in ACC do not support the understanding of the behaviour and limitations of the system. A driving simulator study by *Adell et al. (2008)* provided a comprehensive analysis of combinations of interfaces for supporting safe driving. Participants in that study were most positive about the haptic interface, while the auditory warning signals were not highly appreciated, which may be explained by the nature of the experiment that exposed the participants to a high urgency scenario of avoiding rear-end collisions (*Adell et al., 2008*).

Findings from the aviation field show that the female voice is more difficult to understand in noisy environments (*Nixon et al., 1998a*). It has also been argued that the female voice has the advantage that it stands out more in a predominantly male environment, such as the military (*Noyes, Hellier & Edworthy, 2006*). Differences in speech intelligibility and perceived urgency between male and female voices are generally small and findings have been mixed (e.g., *Arrabito, 2009*; *Edworthy, Hellier & Rivers, 2003*; *Nixon et al., 1998b*). However, it has been found that most people normally use a female voice when using their route navigation device (*Large & Burnett, 2013*). In the present research, respondents were asked to select the types of interfaces they are willing to be guided by during a takeover request. The results of our survey further showed that the female voice is preferred in both English and non-English speaking countries. Thus, our findings reinforce the idea that the overall most preferred way to support the transition of control is an auditory instruction performed with a female voice. Note that determining the language of respondents based on their IP address cannot guarantee accurate results in all cases. In future surveys adding a question prompting for the participant's spoken language may yield more accurate results.

It was found that the participants showed a relatively low level of appreciation of vibratory interfaces, which contrasts with the findings in *Adell et al. (2008)*. This could be due to the fact that only a small number of systems that feature vibratory feedback are available in modern vehicles. A relatively large number of people indicated that they would like to be aided by all four proposed modalities. In addition, a large number of respondents indicated that the combination of a sound signal, a visual message, and a vibration signal would be preferable during takeover requests in highly automated driving. This is a surprising finding as such a combination is not common in current cars. A possible explanation of this finding could be that the respondents misinterpreted the question and instead of indicating their preference for multimodal feedback, expressed their preferences for the types of feedback that can be used separately from each other during takeover requests in highly automated driving. Another limitation of the present study is that we did not vary possible parameters of the feedback signals, including the quality, intensity, timing, and speed of delivery of the takeover requests. Future experimental research could investigate the effects of such parameters.

The existing systems—the PA and FCWS—received favourable ratings, which may not be surprising, since these systems have already been tested and are already available on the market. One limitation in this context is that the participants relied on a narrative description, complemented with a visual illustration; the survey did not contain actual examples of auditory cues. Before the initiation of the survey, it was believed that the proposed FS would be seen as a way to enhance the enjoyment of driving a car through a scenic place. The results showed that the participants were rather sceptical about such a concept: the system was perceived as somewhat annoying, with a mean score of 3.21 to question Q24 on the scale from disagree strongly (1) to agree strongly (5). The proposed FS was not highly rated, possibly because the concept was perceived as a bad idea, because of a lack of previous experience with such system, or because people could not envision it due to the lack of a realistic representation (see also the analysis of the textual comments). It should also be noted that a large proportion of respondents selected the middle option *Neither agree nor disagree,* possibly indicating difficulties with understanding the concept of the proposed system (for studies into middle category endorsement, see *Kulas, Stachowski & Haynes, 2008*; *Kulas & Stachowski, 2009*; *Sturgis, Roberts & Smith, 2012*).

Small effects of age on the acceptance of FAD were previously reported by *Payre, Cestac & Delhomme (2014)*. In the present study, we also observed small age effects regarding the self-reported annoyance of the three proposed systems: younger participants saw the PA and FCWS as more annoying than older respondents did. However, young respondents perceived the FS as less annoying than the older respondents. It is known that younger people are more likely to accept new technologies (*Lee, 2007*; *Tacken et al., 2005*), and thus may be more successful at envisioning such abstract concepts as the FS. A somewhat stronger age effect was observed regarding helpfulness: older respondents found the FCWS more helpful than the younger participants. It is known that young people feel more confident behind the wheel (*Matthews & Moran, 1986*; *Lee et al., 2002*; *Lee, 2007*; *Clarke, Ward & Truman, 2005*), and therefore may think they need less external help than older drivers.

CrowdFlower offers a platform that supports full anonymity of participants. This anonymity may have encouraged respondents to express their thoughts freely, without the fear of being judged by the organizers of the survey. All but the last free-response items required people to enter at least one character. A large number of respondents did not provide meaningful comments. However, a substantial portion of respondents did provide valuable answers, facilitating the understanding of what people think about not only the use of auditory interfaces in future highly and fully automated cars, but also about the concept of automated driving in general. Numerous respondents expressed their concerns about the qualities of current PA and FCWS systems. Some participants suggested that they want to be aided by visual and vibratory feedback in addition to auditory feedback. A number of people indicated the inaccessibility of modern PAs and FCWSs to deaf users. However, current systems also provide haptic and/or visual cues (*BMW, 2013*; *Volvo, 2013b*; *Ford, 2014*; *Honda, 2014*), and so people with a hearing impairment could still benefit from such multimodal feedback. Some respondents were sceptical about the introduction of highly and fully automated vehicles, which may be related to general consumer scepticism about new technologies. Respondents expected that most cars would drive fully automatically by the year 2030 (median value), a result that matches findings in previously published research (*Sommer, 2013*; *De Winter et al., 2015*; *Kyriakidis, Happee & De Winter, 2015*).

The total cost of the study performed by means of a crowdsourced online-based survey was lower than what is offered by companies that conduct similar surveys with help of classic recruitment methods (*De Winter et al., 2015*). A group of people filled in the survey more than once, and we reasoned that their responses ought not to be trusted. Thus, we applied a strict filter, and removed all respondents who filled out the survey more than once. We also excluded all people who had one or more missing items. With appropriate data quality control mechanisms, crowdsourcing is known to be a powerful research tool (*Howe, 2006*; *Kittur, Chi & Suh, 2008*; *Mason & Suri, 2012*; *Crump, McDonnell & Gureckis, 2013*). Nonetheless, as with any self-report questionnaire, the validity of the results is limited to what people can imagine or retrieve from their memory. Furthermore, CrowdFlower respondents are not representative of the entire population of stakeholders of future HAD cars. It is likely that highly automated vehicles will initially be purchased by wealthy people, while projects on CrowdFlower are often completed by people from low-income countries (*Kyriakidis, Happee & De Winter, 2015*).

In conclusion, the present survey study showed that the PA and FCWS were well appreciated by respondents, whereas the proposed future system (FS) was not rated highly. A second conclusion is that the female voice is the most preferred takeover request among the offered options. The scientific community and the automotive industry may be able to use the information gathered in the present survey for the development of automated driving systems, in particular future iterations of parking assistants and forward collision warning systems, as well as for the design of human-machine interfaces for automated driving.

## ACKNOWLEDGEMENTS

We would like to express our special gratitude to Daria Nikulina for designing the illustrations used in the survey.

### Funding

The research presented in this paper is being conducted in the project HFAuto–Human Factors of Automated Driving (PITN-GA-2013-605817). The funders had no role in study design, data collection and analysis, decision to publish, or preparation of the manuscript.

### Grant Disclosures

The following grant information was disclosed by the authors:
HFAuto–Human Factors of Automated Driving: PITN-GA-2013-605817.

### Competing Interests

The authors declare there are no competing interests.

### Author Contributions

- Pavlo Bazilinskyy conceived and designed the experiments, performed the experiments, analyzed the data, contributed reagents/materials/analysis tools, wrote the paper, prepared figures and/or tables, performed the computation work, reviewed drafts of the paper, organised outsourcing of creation of illustrations.
- Joost de Winter conceived and designed the experiments, performed the experiments, analyzed the data, contributed reagents/materials/analysis tools, wrote the paper, prepared figures and/or tables, performed the computation work, reviewed drafts of the paper.

### Human Ethics

The following information was supplied relating to ethical approvals (i.e., approving body and any reference numbers):

The research was approved by the Human Research Ethics Committee (HREC) of the Delft University of Technology.

### Supplemental Information

Supplemental information for this article can be found online at http://dx.doi.org/10.7717/peerj-cs.13#supplemental-information.

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
