# Peer review of "Auditory interfaces in automated driving: an international survey"

_PeerJ Computer Science, doi:10.7717/peerj-cs.13_

## Round 0.1 · original submission · Minor Revisions

The reviewers agreed that the paper was clearly written, and the survey was in general correctly designed and ethically conducted. Two reviewers requested that the aim and conclusions be made clearer, so please give some attention to this; plus of course address the other points raised by the reviewers. Also please remember to address the various formatting issues that have been raised (PeerJ does not perform copyediting).

·

Basic reporting

The paper contains clear background material that serves as a readable introduction to someone unfamiliar with the field. The English language contains a few minor grammatical errors but is perfectly understandable throughout. The structure of the paper is logical and well organised. The figures and tables included are clearly presented and necessary to the arguement of the paper.

The aims of the survey are clearly defined.
The case for the use of audio as a communications mode for in-car systems is well made.
The coverage of the advantages of auditory displays is not comprehensive, but adequate.
Some coverage of the downsides of auditory displays should be added, such as sounds being masked by other sounds, the fact that they are transient leading to them possibly being missed by the driver ...

The description of an earcon is not in line with established auditory display literature (1), and appears somewhat confused with the concept of an auditory icon (1).

In general, a more detailed coverage of relevant previous research on auditory displays (1) would strengthen the theoretical background presented and lay the ground for a more insightful discussion of the options for the design of audio Parking Assistance (PA) and Forward Collision Warning Systems (FCWS).

The paper provides a clear introduction to PA and FCWS and their current limitations.

A figure is given for the number of PA systems currently available but no equivalent is given for FCWSs, though some figures are quoted for a survey of Volvo users.

The details of the survey questionnaire and the results and analysis of these are clearly presented (but see comments on experimental design and validity below).

The references appear correct and complete.

Experimental design

The methods employed would be completely reproducible if desired.
Appropriate ethical clearance has been obtained for the work.

As the authors acknowledge, a driving simulator based examination of the issues covered in this paper would provide more ecologically valid results, though with the disadvantages of substantial extra cost and having a much smaller number of participants.

Participants were asked about preference for the gender of the voice supporting driver takeover messages. The gender is of course only one characteristic of the voice: natural vs. synthesised, quality of reproduction, level of prosedy, speed of delivery and level of driver configurability are at least some other important parameters to be taken into account. Again these could be tested through simulation.

It seems strange that illustrations were used to represent the 3 scenarios, PA, FCWS and Future System (FS), whereas there sole mode of communication would be audio. This seems particularly flawed in the case of FS, as participants are unlikely to have any previous relevant auditory experience to draw upon. This limitation is acknowledged in the Discussion section of the paper. There may have been technical issues in delivering a realistic auditory representation of FS to participants via the Internet, but there doesn't seem to be any good reason why audio samples of typical PA and FCWS systems could not have been provided. I don't know whetehr CrowdFlower supports audio, but if not, this shortfall in the study design surely made it worth considering alternative means of delivering the survey.

The survey questionnaire employed appears generally clear and unambiguous.
It is however hard to know how respondents could answer the question adequately concerning the surround sound of the FS system without having an auditory example to go on.

In the survey, interpreting the scale for frequency of driving (question 4) does not seem to be entirely straightforward, for example, two participants giving the same values of 2, 3 and 4 are likely to interpret what is meant by those figures somewhat differently. Specifying what is meant by more of the values in the scale could have reduced the scope for different interpretations of those values.

The authors analysed for English and non-English speaking countries, it would be interesting to know if there was a deliberate decision to analyse in that way as opposed for example to asking a question about the participants first language, or nationality, as clearly someone's location does not necessarily correlate with their nationality or first language.

The authors have gone to some trouble to remove completed questionnaires they had reason to be doubtful about, i.e. where participants had not read instructions, were under driving age etc. This still left a healthy number of completed surveys (1205).

The large number of neutral responses concerning the FS could be down to the lack of a realistic representation and lack of previous experience of such a system on the part of survey participants.

The evidence of the preference for a female voice, both among male and female participants is compelling.

The lack of support for vibrations as a warning mechanism is interesting.

Validity of the findings

The results of the survey are clearly stated and in general are justified by the data presented.

The conclusions and reflections on the results ar generally reasonable and the limitations of the survey appropriately acknowledged. There is some speculation in the discussion, but it is clear when this occurs and seems completely reasonable given the nature of the survey and its findings.

The contrast between the lack of enthusiasm for vibrations as a message carrier in the survey results contrasts with the driving simulation study by Adell, et al mentioned in the discussion. This could be due to the lack of experience of such systems among the survey participants.

The relatively positive attitudes in the survey to the PA andFCWS audio systems (in comparison with other studies) may in part be due to the lack of audio indicatinghow these would actually sound, but negativity inspired by the sounds of current systems can surely be addressed by better sound design, which in turn could be evaluated using simulations.

The preferences for multimodal feedback in PA and FCWS systems is a valuable finding. I do not agree with the authors that the existence of visual feedbac in existing PAs makes the comments from survey respondents about audio only PAs irrelevant. Considering the arguement in favour of multimodality in this context, there is surely an arguement for PA systems with vibratory output so that hearing impaired drivers benefit from multimodality, particularly in view of the directional nature of the visual sense.

The authors rightly acknowledge the shortcomings of self reporting surveys, and the likely mismatch in demographics between Crowd Flower respondens and the early adopters of Fully Automated Driving (FAD) vehicles. Nevertheless I agree with their overall conclusions about the value of their findings in supporting the near future iterations of PA and FCWS designs.

Additional comments

Overall this is a highly readable and interesting contribution. I do have some significant reservations about some of the methods employed in administering the survey, particularly the use of illustrations and no use of sound samples to represent the PA and FCWS systems. Overall however I believe the findings of the study to be valid and a useful contribution to the design of future PA and FCWS systems.

(1) Hermann, T., Hunt, A., Neuhoff, J. G., editors (2011). The
Sonification Handbook. Logos Publishing House, Berlin,
Germany, chapter 2, Theory of Sonification, available online at http://sonification.de/handbook

Reviewer 2 ·

Basic reporting

I will provide my review as a separate document.

Experimental design

I will provide my review as a separate document.

Validity of the findings

I will provide my review as a separate document.

Annotated reviews are not available for download in order to protect the identity of reviewers who chose to remain anonymous.

Reviewer 3 ·

Basic reporting

This article presents the results of an international survey about auditory interfaces in automated driving.

It is well written and easy to read.
There is however one sentence which does not seem clear to me, lines 169 to 172 : why such system might lead to changes in traffic regulation and warning emergency systems? What might be these changes?

The authors followed a clearly organized structure, with a complete introduction and a description of the method, before presenting the findings and discuss them. However, the paper lacks a proper conclusion to make the finding more visible for the reader.

This article includes a large number of relevant references. The reference format should be modified to respect the style used in PeerJ Computer Science ((Name, year) for in-text citations and beginning with the last name in the reference section). Also, there may be a mistake line 538, supporting that “existing PAs already provide visual feedback” with references 55, 56, 60 dealing with FCWS.

Figures are relevant and well described. Figures 4 and 6 may, however, be difficult to read when printed in black and white. More significantly, some values seem inconsistent between tables 2 and 3. For instance for PAhelp, values are said to be included between 1 and 5 on table 2 but there are only frequencies for answers 1, 2 and 3 on table 3. It is also the case for PApast, FCWSpast and FCWShelp.

Most data have been made available. Only textual comments are lacking, but well described in the document.
Results are clearly described and surely useful for designers or manufacturers of automated vehicles.

Experimental design

Experimental design is clearly described and the investigation rigorously conducted, respecting ethical standards. According to me, it just lacks the description of the FS which was given to the respondents to imagine this auditory display.

The large number of respondents and their diversity provides a broad perspective on the opinion of potential clients of such auditory displays. CrowdFlower respondents might not be representative of the entire population and it could be interesting to compare these findings to wealthier population, maybe more used to such technology (PA and FCWS) and more likely to purchase these vehicles. Nonetheless, this point is well discussed in this paper. It is not compulsory for this study and could be considered as a future development.

Moreover, it might seem delicate to ask people to give their point of view about a technology they only imagine. But this point is also well discussed and the vision of potential users is always instructive. It might have been interesting to compare the findings on PA and FCWS between drivers who are used to these systems and those who are not. Actual users might find it more helpful and, thus, less annoying than non-users.

Validity of the findings

Concerning the data analysis, I wonder if some analyses shouldnot have been extended.
The Spearman correlations given lines 335 and 336 between age and annoyance or helpfulness seem very weak, and might not be statistically significant. Further analyses, like confidence interval or p-values can answer this point.
Similarly, comparisons between male and female respondents or English speakers vs. non-English speakers’ preferences about male or female voice should need further analysis to determine their significance. Confidence intervals or chi square test might answer this point.

Concerning the conclusions, I wonder if the textual comments given lines 426 and 427 reflect more a fear than a real dissatisfaction. Were these respondents common users of FCWS or just imagined than this warning could rapidly become annoying? Are there any changes in the judgments of annoyance and helpfulness between people who use these systems and those who do not?

---

## Round 0.2 · accepted · Accept

The updated manuscript is improved, and you have satisfied most of the points of reviewer feedback. You have not really addressed the request from Reviewer 1 to discuss the downsides of auditory displays. It is implicit in some of the added citations and the very brief mention of masking; the cocktail party effect is not a cure-all!. However I will allow this, given the context.